# Microdialysis and $CO_2$ sensors detect pancreatic ischemia in a porcine model

Kristina Rydenfelt[1,2]*, Runar Strand-Amundsen[3], Rune Horneland[4], Stina Hødnebø[1,2], Gisle Kjøsen[2,5], Søren Erik Pischke[1,2,6], Tor Inge Tønnessen[1,2], Håkon Haugaa[1,7]

1 Division of Emergencies and Critical Care, Department of Anesthesiology, Oslo University Hospital, Oslo, Norway, 2 Institute of Clinical medicine, University of Oslo, Oslo, Norway, 3 Department of Clinical and Biomedical Engineering, Oslo University Hospital, Oslo, Norway, 4 Department of Transplantation Medicine, Section of Transplantation Surgery, Oslo University Hospital, Oslo, Norway, 5 Division of Emergencies and Critical Care, Department of Research & Development, Oslo University Hospital, Oslo, Norway, 6 Department of Immunology, Oslo University Hospital, Oslo, Norway, 7 Lovisenberg Diaconal University College, Oslo, Norway

* kriryd@ous-hf.no

## Abstract

### Background

Pancreatic transplantation is associated with a high rate of early postoperative graft thrombosis. If a thrombosis is detected in time, a potentially graft-saving intervention can be initiated. Current postoperative monitoring lacks tools for early detection of ischemia. The aim of this study was to investigate if microdialysis and tissue $pCO_2$ sensors detect pancreatic ischemia and whether intraparenchymal and organ surface measurements are comparable.

### Methods

In 8 anaesthetized pigs, pairs of lactate monitoring microdialysis catheters and tissue $pCO_2$ sensors were simultaneously inserted into the parenchyma and attached to the surface of the pancreas. Ischemia was induced by sequential arterial and venous occlusions of 45-minute duration, with two-hour reperfusion after each occlusion. Microdialysate was analyzed every 15 minutes. Tissue $pCO_2$ was measured continuously. We investigated how surface and parenchymal measurements correlated and the capability of lactate and $pCO_2$ to discriminate ischemic from non-ischemic periods.

### Results

Ischemia was successfully induced by arterial occlusion in 8 animals and by venous occlusion in 5. During all ischemic episodes, lactate increased with a fold change of 3.2–9.5 (range) in the parenchyma and 1.7–7.6 on the surface. Tissue $pCO_2$ increased with a fold change of 1.6–3.5 in the parenchyma and 1.3–3.0 on the surface. Systemic lactate and $pCO_2$ remained unchanged.

The area under curve (AUC) for lactate was 0.97 (95% confidence interval (CI) 0.93–1.00) for parenchymal and 0.90 (0.83–0.97) for surface ($p<0.001$ for both). For $pCO_2$ the AUC was 0.93 (0.89–0.96) for parenchymal and 0.85 (0.81–0.90) for surface ($p<0.001$ for

**Data Availability Statement:** All relevant data are within the paper and its Supporting Information files.

**Funding:** This work was supported by research grants from South-Eastern Norway Regional Health Authority (Helse Sør-Øst RHF, 2016028), The Norwegian Diabetes Association (36989) and The Norwegian Association of Kidney Patients and Organ Transplanted (36945).

**Competing interests:** We have read the journal's policy and the authors of this manuscript have the following competing interests: Tor Inge Tønnessen is medical advisor, member of the board and shareholder for the IscAlert PtCO2 sensors (IscAlertTM) producing company Sensocure AS. Runar Strand-Amundsen is a researcher at Sensocure AS. No other authors report any conflict of interest.

both). The median correlation coefficients between parenchyma and surface were 0.90 (interquartile range (IQR) 0.77–0.95) for lactate and 0.93 (0.89–0.97) for $pCO_2$.

## Conclusions

Local organ monitoring with microdialysis and tissue $pCO_2$ sensors detect pancreatic ischemia with adequate correlation between surface and parenchymal measurements. Both techniques and locations seem feasible for further development of clinical pancreas monitoring.

## Introduction

In the postoperative course after pancreatic transplantation, graft thrombosis is the most frequent cause of early graft failure and loss, with the majority occurring within the first postoperative week [1,2]. Early graft thrombosis leads to graft loss in about 5–10%, and accounts for almost one third of grafts lost within the first 6 months after transplantation [3,4]. If detected early, the grafts can be salvaged by endovascular or surgical thrombectomy [5,6]. However, detecting a graft thrombosis before the transplant is irreversibly damaged is challenging, since the first clinical sign is often an increase in blood glucose due to an already severely injured pancreas graft. Strategies that have been proposed to decrease the number of graft losses include surveillance programs with daily color Doppler contrast enhanced ultrasound [7], or immediate post-transplant computed tomography angiography [8]. Since the time to diagnosis is crucial to improve graft survival, monitoring techniques that detect ischemia as-close-to real time as possible are needed to accelerate conclusive diagnostic tests [9].

Microdialysis catheters and $pCO_2$ sensors are designed to monitor organs and tissues at a local level with high sensitivity. Microdialysis catheters consist of one inlet and one outlet tube with a semipermeable membrane at the tip, allowing for sampling of substances like glucose, lactate, pyruvate, and glycerol directly from the tissue in question. Analysis is made in a semi-continuous fashion at the bedside and provides important information of the local metabolic state [10]. The method has been used in clinical studies to detect postoperative complications in abdominal organs, e.g. the bowel and the liver [11].

The tissue $pCO_2$ sensor (IscAlert^TM, Sensocure AS, Skoppum, Norway) is based on the principle of $CO_2$ conductivity. The sensor has a gas permeable membrane allowing for $CO_2$ diffusion into a chamber. The $CO_2$ molecules react with de-ionized water and form ions that change the conductivity of the water and the sensor record the conductivity change [12]. Under anaerobic conditions $CO_2$ will increase both due to increased production and impaired blood transport away from the tissues [13]. In a laboratory experiment, these $pCO_2$ sensors have detected ischemia in real-time in hepatic tissue and small intestine [14]. Other techniques using $pCO_2$ measurements to provide information of tissue ischemia in abdominal organs, have been tested on the liver surface [15], and intraperitoneally adjacent to small intestine [16].

Microdialysis catheters positioned in the parenchyma have confirmed pancreatic ischemia in a porcine model in vivo [17], in porcine and human pancreases ex vivo[18], and we have monitored pancreas grafts with microdialysis catheters on the organ surface [19]. There is no clinical experience with intraparenchymal pancreatic tissue monitoring. Unlike hepatic tissue, where small catheters can safely be inserted into the parenchyma [20,21], inserting a catheter into the pancreas graft puts it at risk of graft pancreatitis and exocrine fistulae. An experimental study of the liver has suggested that changes in intraparenchymal metabolites are mirrored

on the organ surface [22]. In these studies, where microdialysis catheters were first used on organ surfaces, conventional circumferential microdialysis probes, designed to be inserted into the tissue, were used on the organ surface. To customize probes for surface sampling, a unidirectional sampling microdialysis probe (OnZurf Probe, Senzime AB, Uppsala, Sweden), was developed and has been tested on small bowel and esophagus [23,24]. How metabolite concentrations on the surface of the pancreas relate to the intraparenchymal is not yet determined.

The objective of this study was to assess if microdialysis and tissue $pCO_2$ sensors placed locally can detect pancreatic ischemia intraparenchymally, and on the organ surface in a porcine model, and if the measurements from the two locations are comparable.

## Materials and methods

All experiments were performed at the Institute of Surgical Research, Oslo University Hospital/The University of Oslo. It was an open label non-recovery experiment lasting approximately 12 hours. The animal protocol was designed to minimize pain and discomfort to the animals and reduce the overall number of animals used. An animal model was needed since safety with intraparenchymal pancreas monitoring is uncertain. The porcine model was chosen since there is a wealth of experience and global acceptance of translational porcine models. We used CE-marked human microdialysis probes and tissue $pCO_2$ sensors that are currently used in clinical studies, and in the process of receiving a CE marking, that could rapidly be adapted to a clinical setting.

### Ethics

The study was approved by the Norwegian Food and Safety Authority (FOTS ID 7447 and 15378) and complied with the Norwegian regulations on the use of animals in experiments (LOV-2009-06-19-97) and the EU Directive on the protection of animals used for scientific purposes (2010/63/EU). The ARRIVE guidelines checklist (https://arriveguidelines.org/) was used and detailed information about the animals and the animal handling can be found in S1 Table.

### Animals

In total 11 Norwegian landrace pigs (*Sus scrofa domesticus*) were used, two of which served as pilots to develop the surgical model. One animal was euthanized before start of the experimental protocol due to irreversible ischemia in the pancreas during the dissection. Accordingly, 8 animals (4 males/4 females, weight; 45.5–51.5 kg) were included in the study. Anesthesia was induced by ketamine 20 mg/kg, atropine 0.02 mg/kg and azaperone 3 mg/kg and maintained with 1–2% isoflurane and a morphine infusion of 1–2 mg/kg/h. The animals underwent a surgical tracheotomy. A central venous line was established in the internal jugular vein and an arterial line in the internal carotid artery. A crystalloid infusion and a glucose infusion were continuously administered and adjusted to keep mean arterial pressure > 50 mmHg and blood glucose between 5–8 mmol/L. Vasopressors were not used as these could have influenced local tissue perfusion. At the end of the procedure, the animals were euthanized with pentobarbital (1000 mg), morphine (50–100 mg) and potassium chloride (70–100 mmol).

### Laparotomy

All abdominal surgical procedures were performed as midline laparotomies by the same surgical team. In the four first animals, the pancreas was divided to isolate the corpus and the cauda

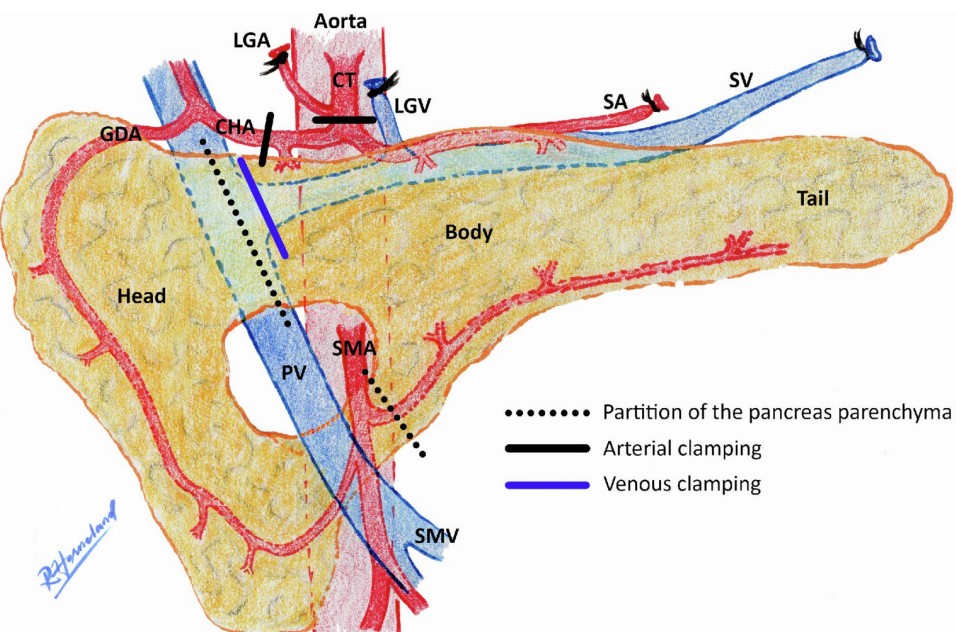

**Fig 1. Outline of the pig pancreas anatomy.** Sites for pancreas parenchyma partition and vessel clamping marked. The tail was used for measurements. CHA, common hepatic artery; CT, coeliac trunk; GDA, gastroduodenal artery; LGA, left gastric artery; LGV, left gastric vein; PV, portal vein; SA, splenic artery; SMA, superior mesenteric artery; SMV, superior mesenteric vein; SV, splenic vein.

on each side of the portal vein, aiming to keep normal circulation in the caput while performing ischemic experiments on the cauda. However, clamping of the vessels to the cauda also induced ischemia in the caput in the four first animals, and these measurements could therefore not serve as non-ischemic controls and were discarded from the study. Measurements in the caput were subsequently abandoned. Instead, the baseline measurements, before occlusion, were used as non-ischemic reference values. We exposed the coeliac trunk and the proper hepatic artery for arterial clamping, and the splenic vein for venous clamping. In addition, the left gastric artery was ligated (Fig 1). As clamping of the splenic vein failed to achieve complete outflow obstruction in animal number 5, we attempted to dissect all surrounding pancreatic tissue in the following animals with the aim of excluding small venous collaterals and to isolate the splenic vein as the only venous outflow from the cauda.

## Microdialysis catheters and tissue $pCO_2$ sensors

We used two catheters/sensors of every type in the parenchyma and two on the surface to assure measurements from every location, except for one animal with no and three animals with only one unidirectional sampling microdialysis and two animals with only one surface $pCO_2$ sensor. Number of catheters/sensors placed in the parenchyma and on the surface of every pig is displayed in S2 Table and a picture of the set-up in S1 Fig. Circumferential sampling microdialysis catheters with a 30 mm membrane, a pore size of 100 kDa, and an outer diameter 0.6 mm (Custommade MDialysis-65 catheter, MDialysis AB, Stockholm, Sweden, based on MDialysis-61, https://www.mdialysis.com/product/61-microdialysis-catheter/) were inserted into the pancreatic parenchyma using splittable introducer needles and placed onto the pancreatic capsule and sutured to the surface of the pancreas. Tissue $pCO_2$ sensors (IscAlert™, Sensocure AS, Skoppum, Norway, https://sensocure.no/products/), with diameter 0.7 mm, were inserted into the pancreatic tissue and sutured on the surface in the same manner as

with the microdialysis catheters. From the second pig onwards, in addition to the conventional circumferential sampling microdialysis catheters, we used unidirectional sampling microdialysis catheters with a 15 mm membrane, surface area of 0.2cm$^2$ and a pore size of 10 kDa, designed to collect samples from the organ surface only (OnZurf Probe, Senzime AB, Uppsala, Sweden, https://senzime.com/products/product-pipeline/onzurf). For all affixments we used 6–0 sutures. All microdialysis catheters were perfused with 6% hydroxyethyl starch (Voluven® 60mg/ml, Fresenius Kabi AS, Halden, Norway) at a rate of 1 μL/min using microinjection pumps (107 Microdialysis pump, M Dialysis AB, Stockholm, Sweden). The microdialysate was collected in microvials (MDialysis AB, Stockholm, Sweden) and analyzed for glucose, pyruvate, lactate and glycerol in a microdialysis analyzer (ISCUS$^{flex}$, MDialysis AB, Stockholm, Sweden).

## Experimental protocol

Following surgery and the insertion of the catheters, we implemented a 60-minute stabilization period before clamping the coeliac trunk. To avoid retrograde flow to the caudal part of the pancreas (where the measurements were done) we also clamped the common hepatic artery. The coeliac trunk and the proper hepatic artery were clamped for 45 minutes followed by a reperfusion period of 120 minutes. The splenic vein was then clamped for 45 minutes at its entrance into the portal vein, followed by a 120-minute reperfusion period (Fig 2).

## Measurements

Hemodynamic and respiratory parameters (heart rate, blood pressure, central venous pressure, and peripheral oxygen saturation) were monitored continuously and recorded every 15 minutes throughout the experiment. Arterial (the internal carotid artery) and venous (the internal jugular vein) blood gases were analyzed every 30 minutes during the experiment (ABL 825 FLEX, Copenhagen, Denmark). The microdialysis samples were analyzed every 15 minutes, and the $pCO_2$ values were recorded 1–2 times per second and immediately displayed graphically on a computer screen. Only the $pCO_2$ values occurring on every 5-minute mark were used for calculations.

## Statistical analysis

The sample size of eleven animals were chosen based on our experience from earlier studies and not from sample size calculation. Based on data from a previous study [14], we anticipated a more than 100% increase in the main target variables, lactate and $pCO_2$ following total vessel occlusion. A power calculation, assuming a minimum lactate increase of 50%, a power of 80% and a significance level of 5%, suggested a minimal sample size of 4 animals. Since this

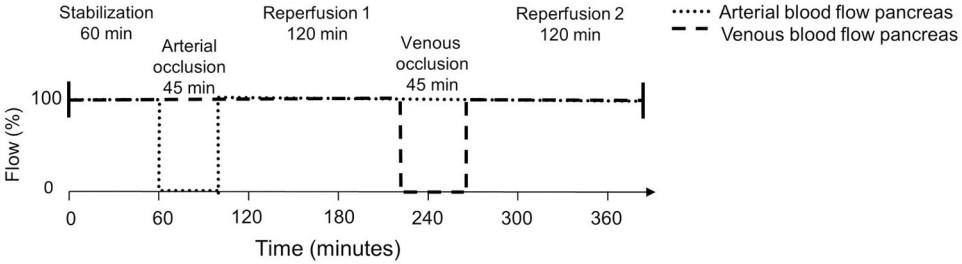

**Fig 2. Experimental protocol.** Starting after surgery and placement of monitoring catheters, demonstrates the timeline for blood flow occlusions to the pancreas with following periods of reperfusion.

experimental model had not been employed before, we needed to account both for animals used as pilots to develop the model and for missing values when we determined the sample size. The primary outcomes were ischemia induced changes in metabolites and $pCO_2$ measured at the organ level.

The data are presented as medians with interquartile ranges (IQR) for all variables. All variables apart from blood pressures were non-normally distributed according to histogram and Shapiro Wilk's test. All vital and laboratory parameters obtained immediately before the occlusions were compared to values right before reperfusion with the Wilcoxon signed rank test. We compared the microdialysis parameters and $pCO_2$ values obtained right before occlusions with the most divergent values after occlusions using the Wilcoxon signed rank test. Whenever two microdialysis catheters or $pCO_2$ sensors had the same location (parenchyma or surface), the mean values from the two catheters were used. Pearson correlation coefficients were calculated for every pair of catheters or sensors placed in the same location.

Pearson correlation coefficients were also calculated for each animal between the parenchymal and the surface values. Receiver operating characteristics (ROC) analyses were performed for lactate (circumferential and unidirectional sampling catheters) and $pCO_2$, both for parenchymal and surface measurements, to obtain tentative cutoff values and to calculate sensitivity and specificity for ischemia. In the ROC analysis, all baseline values before arterial occlusion and stable values between the first reperfusion and the venous occlusion were used as control, and all values during arterial and venous clamping were used as ischemia/event. The cutoff values for detecting ischemia were determined from the upper left corner of the ROC-curves.

IBM SPSS Statistics for Windows, Version 25.0 (Armonk, NY: IBM Corp) was used for the ROC analysis. Stata Statistical Software (Release 15, StataCorp.2017, College Station, TX: StataCorp LLC) was used for Wilcoxon signed rank test and Pearson correlations. A *p*-value of <0.05 was considered statistically significant. The statistical analyses of this study were reviewed by Øivind Skare, statistician and researcher at the National Institute of Occupational Health Oslo, Norway.

## Results

Arterial clamping induced pancreatic ischemia in all eight animals. In three of the animals, clamping of the splenic vein did neither lead to any increased lactate or tissue $pCO_2$, nor to any discolored or edematous pancreas, and we interpreted it as we failed to induce complete venous outflow occlusion in these animals. Thus results of 8 animals were included from the part of the experiment with arterial occlusion and 5 animals were included from the part of the experiment with venous occlusion. In one animal, data from surface catheters are based on one catheter only during arterial occlusion and no data was obtained during venous occlusion due to malfunction of sensors. The numbers of animals with successful measurements for the different sensor types during arterial and venous occlusions are displayed in Table 1. The number of catheters/sensors with successful measurements from each location are displayed in S2 Table. The vital parameters and blood gas values including $pCO_2$ were stable throughout the experiment (Table 2).

### Microdialysis

**Circumferential sampling.** Lactate measured with circumferential sampling microdialysis probes increased in the pancreas parenchyma, and on the organ surface, in response to pancreatic ischemia, without changes in systemic arterial or venous blood lactate (Fig 3A). Following arterial occlusion, the lactate values increased from baseline with a median (IQR) fold change of 6.2 (4.2–7.2) in the parenchyma (p = 0.008), and 3.0 (2.5–3.8) on the surface

**Table 1. Number of pigs with obtained measurements for the different sensor types.**

| | MICRODIALYSIS | | | PCO$_2$-SENSOR | |
|---|---|---|---|---|---|
| | **Parenchyma (circum-ferential)** | **Surface (circum-ferential)** | **Surface (uni-directional)** | **Parenchyma** | **Surface** |
| Arterial occlusions | 8 | 8 | 7 | 8 | 8 |
| (n = 8) | | | | | |
| Venous occlusions | 5 | 4 | 4 | 5 | 5 |
| (n = 5) | | | | | |

(p = 0.008). The lactate following venous occlusion increased with a fold change of 5.1 (3.9–5.4) from baseline in the parenchyma (p = 0.06), and 2.7 (1.8–3.8) on the surface (p = 0.13). After arterial and venous clamping, glucose decreased more on the surface than in the parenchyma. After the arterial occlusion, pyruvate decreased, and lactate-to-pyruvate (L/P)-ratio and glycerol increased for both catheter locations. After the venous occlusion, the same tendencies were demonstrated (Fig 3B). The median Pearson correlation coefficient (R) between circumferential sampling parenchymal and surface catheter values for lactate for all pigs was 0.90 (0.77–0.95). The correlations of the individual animals are presented in Table 3. ROC analysis demonstrated that parenchymal lactate discriminated ischemia from baseline with an area under curve (AUC) of 0.97 (95%CI 0.93–1.0, p<0.001) and with a cutoff value of 2.3 mmol/L, sensitivity was 94% and specificity 95%. For surface lactate the AUC was 0.90 (0.83–0.97, p<0.001) and with a cutoff value of 3.7 mmol/L sensitivity was 83% and specificity 81% (ROC curve in S2 Fig). Correlation coefficient (R) between pairs of catheters placed in parenchyma and surface are displayed in S3 Table.

**Unidirectional sampling.** Surface lactate measured with the unidirectional sampling catheters increased with a fold change of 4.0 (3.5–4.9) (p = 0.016) after arterial occlusion and 2.4 (2.1–2.9) (p = 0.13) after venous occlusion. Glucose, pyruvate and lactate-to-pyruvate (L/P)-ratios followed the patterns of the circumferential sampling microdialysis catheters. The correlation coefficient (R) between parenchymal circumferential sampling microdialysis catheters and unidirectional sampling surface catheters, was 0.68 (0.63–0.75) (individual animals presented in Table 3). The AUC for the unidirectional surface sampling catheter was 0.81

**Table 2. Systemic vital parameters and laboratory values before and after 45 min of vessel occlusion.**

| | Before arterial occlusion | At 45 min with arterial occlusion | p-value | Before venous occlusion | At 45 min with venous occlusion | p-value |
|---|---|---|---|---|---|---|
| **MAP (mmHg)** | 69 (65–71) | 66 (62–72) | 0.74 | 63(59–68) | 61(53–69) | 0.50 |
| **Heart rate (beats/min)** | 115 (96–132) | 110 (106–144) | 0.11 | 103(95–128) | 111(98–153) | 0.10 |
| **SpO$_2$ (%)** | 96 (94–99) | 96 (95–99) | 0.59 | 96(93–98) | 97(96–98) | 0.84 |
| **Temperature (°C)** | 39.3 (38.2–39.9) | 39.3 (38.0–39.8) | 0.50 | 39.1(38.8–40.5) | 39.3(38.8–40.4) | 0.60 |
| **v-Hemoglobin (g/dL)** | 10.1 (9.2–11.0) | 10.1 (9.1–11.1) | >0.9 | 10.0(8.6–10.6) | 9.3(8.6–11.7) | 0.78 |
| **v-Glucose (mmol/L)** | 5.7 (4.4–6.6) | 5.4 (3.8–5.9) | 0.06 | 4.6(4.1–5.6) | 4.8(4.2–5.6) | 0.87 |
| **a-Lactate (mmol/L)** | 0.7 (0.5–0.8) | 0.6 (0.5–0.8) | 0.67 | 0.7(0.6–0,7) | 0.7(0.5–0.7) | 0.50 |
| **v-Lactate (mmol/L)** | 0.9 (0.5–1.0) | 0.9 (0.8–1.0) | 0.18 | 0.8(0.6–0.8) | 0.9(0.6–0.9) | 0.30 |
| **v-pCO$_2$ (kPa)** | 7.6 (7.1–8.0) | 7.5 (7.2–8.2) | 0.38 | 7.2(6.7–8.1) | 7.4(7.1–7.9) | 0.08 |
| **v-pH** | 7.31 (7.29–7.38) | 7.36 (7.32–7.37) | 0.67 | 7.34(7.32–7.35) | 7.32(7.28–7.32) | 0.31 |

Data presented as median and interquartile range. Values before occlusion have been compared with values at 45 min with vessel occlusion with Wilcoxon signed rank test. No significant changes were observed after 45 minutes of vessel occlusions for any of the systemic parameters measured. MAP, mean arterial pressure; v, venous; SpO2, Saturation of Peripheral Oxygen.

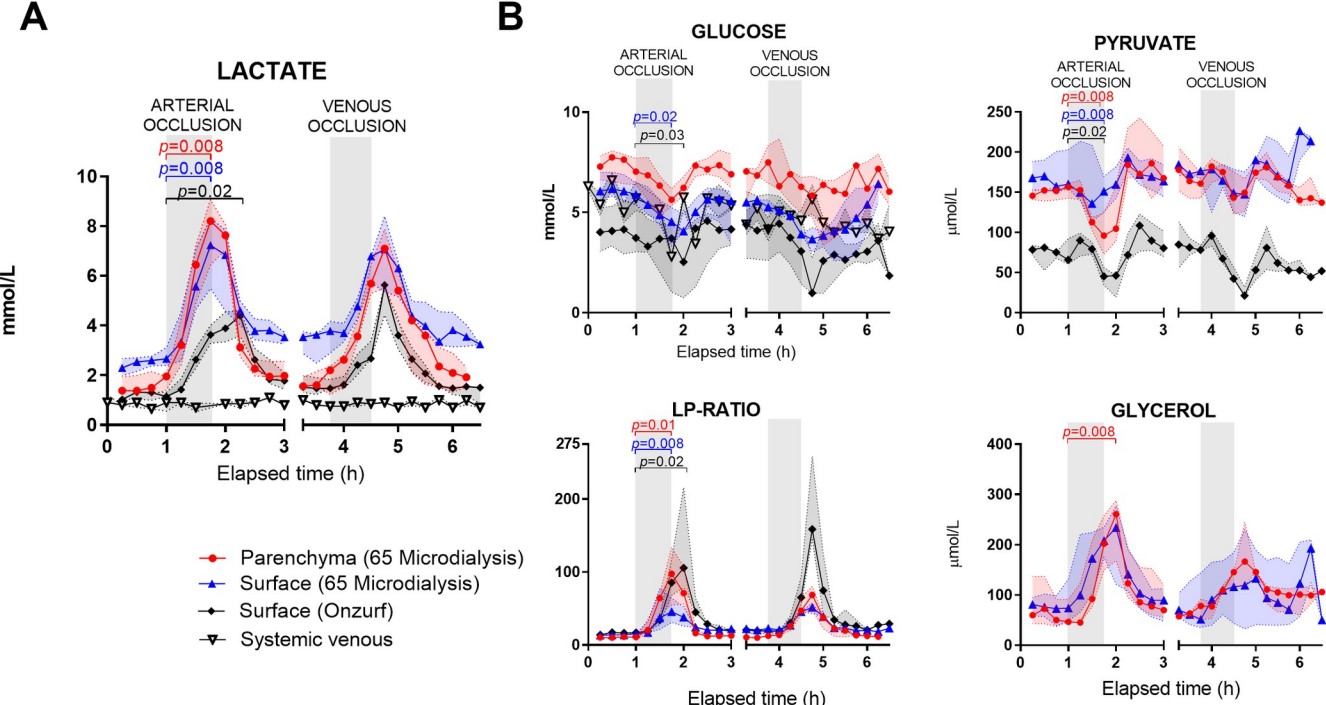

**Fig 3. Microdialysis parameters during arterial and venous occlusions in the porcine pancreas.** In addition, lactate and glucose were measured from systemic venous blood gases. (A) Increase in parenchymal and surface lactate after arterial and venous occlusions. Systemic lactate remained unaffected. (B) Decrease in glucose and pyruvate and increase in lactate-to-pyruvate (L/P)-ratios and glycerol, in the parenchyma and on the surface after arterial and venous occlusions. Circumferential sampling catheter refers to MDialysis 65 and unidirectional sampling catheter to OnZurf. Values presented as median and interquartile range. Maximal changes after occlusions were compared to values right before occlusions with Wilcoxon signed rank test. Significance level $p < 0.05$. Number of pigs included presented in Table 1. L/P-ratio, lactate-to-pyruvate ratio.

(95% CI 0.72–0.91, $p<0.001$) and with a cutoff value of 1.7 mmol/L sensitivity was 75%, specificity 84% (ROC curve in S2 Fig). Correlation coefficients (R) were calculated between pairs of unidirectional sampling for four pigs (S3 Table).

## $PCO_2$

$PCO_2$ increased following both arterial and venous occlusions, while the systemic venous $pCO_2$ remained unchanged (Fig 4).

**Table 3. Pearson correlation coefficients (R) between surface and parenchymal measurements.**

| | LACTATE | | | | | $CO_2$ |
|---|---|---|---|---|---|---|
| | Surface (circumferential) (circumferential) | | Surface (unidirectional) | | | Surface |
| | Parenchyma (circumferential) | | Parenchyma (circumferential) | | | Parenchyma |
| Pig nr | R | $p$-value | R | $p$-value | R | $p$-value |
| 1 | 0.96 | <0.001 | - | - | 0.97 | <0.001 |
| 2 | 0.92 | <0.001 | 0.73 | 0.001 | 0.93 | <0.001 |
| 3 | 0.85 | <0.001 | 0.68 | <0.001 | 0.88 | <0.001 |
| 4 | 0.44 | 0.026 | 0.68 | <0.001 | 0.98 | <0.001 |
| 5 | 0.51 | 0.011 | 0.57 | 0.003 | 0.56 | <0.001 |
| 6 | 0.87 | <0.001 | 0.76 | <0.001 | 0.89 | <0.001 |
| 7 | 0.95 | <0.001 | 0.47 | 0.017 | 0.92 | <0.001 |
| 8 | 0.97 | <0.001 | 0.85 | <0.001 | 0.95 | <0.001 |

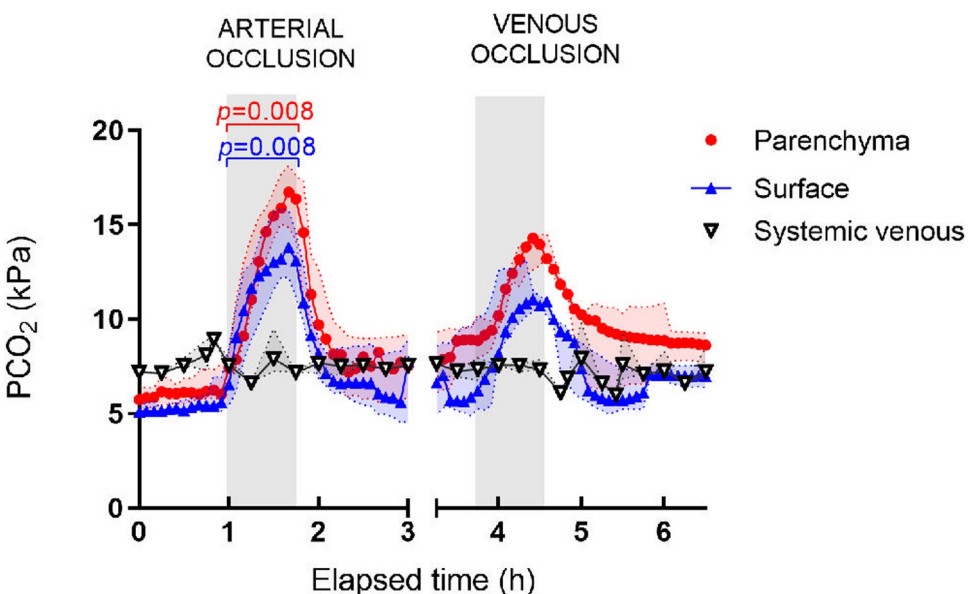

**Fig 4. $PCO_2$ measurements during arterial and venous occlusions in the porcine pancreas.** Increase in parenchymal and surface $pCO_2$ after arterial and venous occlusions. Systemic $pCO_2$ remained unaffected. Values are presented as median and interquartile range. Maximal increases after occlusions were compared to values right before occlusions with Wilcoxon signed rank test. Significance level $p < 0.05$.

After arterial occlusion, the $pCO_2$ values increased rapidly with a median (IQR) fold change of 2.4 (2.2–2.9) from baseline in the parenchyma ($p = 0.0078$) and 2.4 (2.0–2.5) on the surface ($p = 0.0078$). During venous occlusion, the $pCO_2$ increased with a fold change of 1.7 (1.7–1.8) from baseline in the parenchyma ($p = 0.0625$) and 1.7 (1.4–1.9) on the surface ($p = 0.0625$).

The Pearson correlations between surface and parenchymal values of $pCO_2$ were significant for all animals (Table 3) with a median R of 0.93 (0.89–0.97). ROC-analysis demonstrated that $pCO_2$ discriminated ischemia from baseline in the parenchyma with an AUC of 0.93 (95% CI 0.89–0.96), p<0.001), and with a cutoff value of 9.0 kPa sensitivity was 85%, and specificity 82%. The surface $pCO_2$ showed an AUC of 0.85 (0.81–0.90, p<0.001) and with a cutoff value of 6.8 kPa sensitivity was 80%, and specificity 77%. ROC curves for $pCO_2$ are presented in S3 Fig. Correlation coefficients between pairs of $pCO_2$ sensors placed in the parenchyma (n = 8) and on the surface (n = 2) are displayed in S4 Table.

## Discussion

In this experimental porcine model we detected local pancreatic ischemia using real time monitoring with microdialysis catheters and tissue $pCO_2$ sensors, without concurrent changes in systemic hemodynamic or laboratory parameters. In response to ischemia, we observed increased lactate and tissue $pCO_2$. These metabolic changes were detectable on the pancreas surface and correlated well with the parenchymal values.

The successful biosensor based local detection of pancreatic ischemia, together with the absence of systemic markers of ischemia, supports the previous findings, and emphasizes the need for local tissue monitoring [17,19].

We used two different types of microdialysis catheters for the surface measurements: a traditional circumferential sampling microdialysis catheter (MDialysis-65) and a novel unidirectional sampling microdialysis catheter (OnZurf). The correlation between lactate on the pancreas surface and in the parenchyma, as observed with the circumferential sampling

microdialysis catheters, is in line with a previous experimental study on liver where surface lactate concentrations reflected the parenchymal [22]. In this previous study, the highest lactate values were detected on the organ surface during the ischemic episodes. In contrast, we measured higher peak lactate concentrations in the parenchyma during arterial ischemia, but similarly higher surface than parenchymal values during the rest of the experiment. Higher lactate values is to be expected in the parenchyma, since lactate is produced by the ischemic tissue. The higher surface lactate values, can be explained by lactate accumulation in the abdominal fluid, detected by the circumferential catheters that allow sampling from the surroundings. Probably due to the same reason, we observed that surface lactate measured with the circumferential sampling catheters did not return to baseline after the first reperfusion. Conversely, lactate measured with unidirectional catheters returned to baseline. This could support the theory that these catheters are less influenced by the surroundings and that unidirectional sampling catheters, being encapsulated on three sides only and sampling towards the organ surface, is an advantageous alternative for organ surface monitoring.

In our study, the relative lactate increases detected with unidirectional and circumferential sampling catheters placed on the organ surface were comparable. However, the absolute values measured with the unidirectional sampling catheters were lower. A possible explanation for this observation, other than less sampling from the surroundings, could be differences in recovery rates due to different catheter membrane properties. The unidirectional sampling OnZurf catheter has a shorter membrane length (15 vs 30 mm) and smaller pore size membrane (10 vs 100 kDa) compared to the circumferential sampling MDialysis-65 catheter. With the same flow rate as in this study, and with catheters comparable to the MDialysis-65 catheter, lactate recovery came out close to a 100% [25], whereas for OnZurf, lactate recovery was reported to 75% by the manufacturer (Senzime, unpublished data).

Besides the lactate increase, the decrease in glucose and pyruvate and increase in L/P-ratio confirmed metabolic changes seen previously in the pancreas in response to ischemia [17]. In addition, we demonstrated that the metabolite changes were detectable on the pancreas surface with both circumferential and unidirectional sampling catheters. This agrees with results from studies investigating surface monitoring of other organs [22–24]. In our study, an increase in pancreas surface glycerol sampled with circumferential catheters mirrored the increase in the parenchyma. Due to a glycerol coating of the catheter, the unidirectional sampling OnZurf catheter is unable to measure glycerol.

Tissue $pCO_2$ increased after pancreatic ischemia and confirms earlier results where the $pCO_2$ sensors detected organ ischemia in liver and intestinal tissue [14]. Moreover, we demonstrated high correlations between pancreas surface and parenchymal measurements in all animals but one. This was the first time the IscAlert™ tissue $pCO_2$ sensors were used for organ surface monitoring. The absolute $pCO_2$ values on the surface were overall lower compared to the parenchymal. Furthermore, larger variations were observed between the surface sensors than between the parenchymal sensors. We speculate that this may be due to that some of the surface sensors were positioned under the organ surrounded by fluid, and others on the upper side of the organ occasionally towards air. The reasons for more failures with the surface sensors were either that the tip of the catheter was dislodged, and ended up in the air, or a problem with leakage from the catheter and drifting.

In this experimental setting, lactate and $pCO_2$ values, both in the parenchyma and on the surface, differed between ischemic and non-ischemic tissue. Sensitivity and specificity were slightly improved with the parenchymal measurements for both monitoring techniques.

Parenchymal monitoring is probably more reliable due to little or no influence from surrounding conditions. On one hand, surface sensors can be affected by surrounding conditions such as a hematoma, accumulated lactate in abdominal fluid or a local infection [19]. On the

other hand, surface monitoring is less invasive and presumptively safer. It is unknown if insertion of sensors with diameter < 1mm into the parenchyma would cause complications and it cannot be completely ruled out that fixation of sensors on the organ surface does not cause any damage. Monitoring of human pancreas with sensors inserted into the organ or attached onto the organ, needs to be preceded by a safety assessment study. One option could be monitoring animals for several days and a final examination of the pancreas histology regarding inflammation and leakages. However, since tissue ischemia was adequately detected by surface sensors, a reasonable next step would be to develop and improve surface monitoring. Unidirectional microdialysis is inviting, as presumed less influenced by the surroundings, but the current affixment to the organ surface needs to be evaluated from a safety perspective. Future studies are needed to explore if surface tissue $pCO_2$ sensors potentially can be influenced by e.g. a surrounding hematoma or a pancreatic leakage.

Our study has some limitations. The anatomy of the pig pancreas has a more complex blood supply compared to that of the human pancreas. We experienced challenges achieving reproducible ischemia during venous clamping and got fewer results from this part of the study. Furthermore, we had difficulties establishing a non-ischemic reference part of the pancreas. To ascertain ischemia in the caudal part of the pancreas we had to clamp the coeliac trunk. As a result, we also induced ischemia in the head of the pancreas and concluded that the head was dependent on blood supply from the coeliac trunk. If a smaller and more distal part of the pancreatic tail was used, it is possible that clamping of only the splenic artery would have been sufficient. Another way to keep the head non-ischemic could be to place a bypass from the aorta to the common hepatic artery and maintain blood supply to the head via the gastroduodenal artery. Since we were unable to measure from a non-ischemic part of the pancreas, we used the baseline values as reference and each animal served as its own control.

We aimed to examine, but not compare, both arterial and venous ischemia. All venous occlusions were performed after the arterial ischemic episodes, which may have altered the ischemic tissue response. Nevertheless, we chose this sequence of ischemia since we assumed that venous occlusion would induce more edema and organ damage compared to arterial occlusion [26].

## Conclusion

In conclusion, organ close microdialysis and tissue $pCO_2$ sensors detected and demonstrated dynamic and timely indications of pancreas tissue ischemia, not reflected with the current practices of systemic arterial and venous lactate monitoring. Both techniques are promising for rapid detection of postoperative complications leading to organ ischemia. Since lactate and $pCO_2$ were detected both in the parenchyma and on the surface of the pancreas, with adequate correlation between the locations, surface probes is a feasible option for further development of pancreatic monitoring in a clinical setting.

## Supporting information

**S1 Fig. Pig pancreas with microdialysis catheters and tissue $pCO_2$ sensors.** In addition the strings used for arterial and venous clamping are marked.
(TIF)

**S2 Fig. Receiver operating characteristic (ROC) to discriminate ischemia from normal conditions (lactate).** Lactate curves shown for parenchymal and surface values measured with two different microdialysis catheters, the circumferentially sampling MDialysis-65 catheter

and the unidirectional sampling OnZurf catheter.
(TIF)

**S3 Fig. Receiver operating characteristic (ROC) to discriminate ischemia from normal conditions (CO$_2$).** PCO$_2$ curves from parenchymal and surface values measured with the tissue pCO$_2$ sensors.
(TIF)

**S1 Table. Detailed information about pigs (*Sus scrofa domesticus*) used in accord with ARRIVE guidelines.**
(DOCX)

**S2 Table. Number of microdialysis catheters with results from each location/number of catheters placed.**
(DOCX)

**S3 Table. Correlation coefficients (R) for lactate between microdialysis catheters placed in the same location (parenchyma or surface).**
(DOCX)

**S4 Table. Correlation coefficients (R) for pCO$_2$ between tissue pCO$_2$ sensors placed in the same location (parenchyma or surface).**
(DOCX)

**S1 Dataset. Underlying data for all tables and figures.**
(XLSX)

## Acknowledgments

We would like to thank the operational nurses at the Institute for surgical research, and research and development engineer Rune Veddegjerde at Sensocure for excellent assistance during the experiments. Senior engineer Camilla Schjalm at the Department for Immunology for practical advice on laboratory analyses and senior engineer Anders Johnsen at the medical technical staff for invaluable help with repairs of equipment. Dr Itai Schalit for statistical discussions, and statistician and researcher, PhD, Øivind Skare at the National Institute of Occupational Health for statistical review.

## Author Contributions

**Conceptualization:** Kristina Rydenfelt, Runar Strand-Amundsen, Rune Horneland, Tor Inge Tønnessen, Håkon Haugaa.

**Data curation:** Kristina Rydenfelt, Håkon Haugaa.

**Formal analysis:** Kristina Rydenfelt, Runar Strand-Amundsen, Søren Erik Pischke, Håkon Haugaa.

**Funding acquisition:** Tor Inge Tønnessen, Håkon Haugaa.

**Investigation:** Kristina Rydenfelt, Runar Strand-Amundsen, Rune Horneland, Stina Hødnebø, Tor Inge Tønnessen, Håkon Haugaa.

**Methodology:** Kristina Rydenfelt, Runar Strand-Amundsen, Rune Horneland, Søren Erik Pischke, Tor Inge Tønnessen, Håkon Haugaa.

**Project administration:** Kristina Rydenfelt, Håkon Haugaa.

**Resources:** Kristina Rydenfelt, Runar Strand-Amundsen, Rune Horneland, Tor Inge Tønnessen, Håkon Haugaa.

**Supervision:** Søren Erik Pischke, Tor Inge Tønnessen, Håkon Haugaa.

**Validation:** Kristina Rydenfelt, Runar Strand-Amundsen, Rune Horneland, Stina Hødnebø, Søren Erik Pischke, Tor Inge Tønnessen, Håkon Haugaa.

**Visualization:** Kristina Rydenfelt, Rune Horneland, Søren Erik Pischke, Håkon Haugaa.

**Writing – original draft:** Kristina Rydenfelt, Håkon Haugaa.

**Writing – review & editing:** Kristina Rydenfelt, Runar Strand-Amundsen, Rune Horneland, Stina Hødnebø, Gisle Kjøsen, Søren Erik Pischke, Tor Inge Tønnessen, Håkon Haugaa.

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
