## [Decision Letter · Decision Letter 0]

8 Nov 2021

PONE-D-21-33061Microdialysis and CO2-sensors detect pancreatic ischemia in a porcine modelPLOS ONE

Dear Dr. Rydenfelt,

Thank you for submitting your manuscript to PLOS ONE. After careful consideration, we feel that it has merit but does not fully meet PLOS ONE’s publication criteria as it currently stands. Therefore, we invite you to submit a revised version of the manuscript that addresses the points raised during the review process.

ACADEMIC EDITOR: I agree with the 2 expert reviewers that this is an interesting, novel and relevant paper. The reviewers have commented in great detail and in a very constructive manner below about how the MS can further improve for it to become publishable. I particularly like the suggestion by one of the reviewers to underline that microdialysis + pCO2 both gave more sensitive, more timely, and truly dynamic indications of tissue ischaemia than the current clinical practices of venous and arterial lactate monitoring. Furthermore, both reviewers think the conclusions can be expanded a bit. Pls revise the MS accordingly in a point-by-point fashion and with the greatest attention to details pls.

We look forward to receiving your revised manuscript.

Kind regards,

Frank JMF Dor, M.D., Ph.D., FEBS, FRCS

Academic Editor

PLOS ONE

Journal Requirements:

"I have read the journal's policy and the authors of this manuscript have the following competing interests: Tor Inge Tønnessen is a medical advisor, member of the board and stock holder for the IscAlert PtCO2 sensors (IscAlertTM) producing company Sensocure AS. Runar Strand-Amundsen is a researcher at Sensocure AS. No other authours report any conflict of interest. "

Reviewers' comments:

Reviewer's Responses to Questions

**Comments to the Author**

1. Is the manuscript technically sound, and do the data support the conclusions?

Reviewer #1: Yes

Reviewer #2: Yes

2. Has the statistical analysis been performed appropriately and rigorously? 

Reviewer #1: I Don't Know

Reviewer #2: Yes

3. Have the authors made all data underlying the findings in their manuscript fully available?

Reviewer #1: Yes

Reviewer #2: Yes

4. Is the manuscript presented in an intelligible fashion and written in standard English?

Reviewer #1: Yes

Reviewer #2: Yes

5. Review Comments to the Author

Reviewer #1: This reviewer has no concerns about research ethics or publication ethics.

It is a very interesting paper and adds to the literature around real-time microdialysis and gas sampling in organ transplantation.

I think the team have missed a potentially impactful aspect from their discussion, as this paper introduces the idea that microdialysis + pCO2 both gave more sensitive, more timely, and truly dynamic indications of tissue ischemia than the current clinical practices of venous and arterial lactate monitoring.

Specific comments by section:

Background

Pancreatic transplantation instead of pancreas transplantation

No need for hyphens in tissue-pCO2-sensors, just tissue pCO2 sensors. Same throughout manuscript: pCO2-sensors can just be pCO2 sensors.

Methods

Unclear which were inserted into parenchyma and which attached to surface of pancreas. Currently reads as though the MD catheter was inserted and the tissue pCO2 sensor was attached to the surface. Maybe “In 8 anaesthetized pigs, pairs of lactate monitoring microdialysis catheters and tissue pCO2 sensors were simultaneously inserted into the parenchyma and attached to the surface of the pancreas.”?

No need for hyphens in tissue-pCO2-sensors, just tissue pCO2 sensors

PCO2 was measured…-> Tissue pCO2 was measured…(no capital letter P)

Results

PCO2 increased -> Tissue pCO2 increased

Conclusion -> Conclusions

pCO2-sensors -> tissue pCO2

Introduction

Pancreas transplantation -> pancreatic transplantation

Would be nice to give the percentage lost to thrombosis – it’s shockingly high

The microdialysis catheters -> Microdialysis catheters (talking about the general item)

Need manufacturer of pCO2 sensor

In addition to ref. 15, consider https://doi.org/10.1039/C8AY01807C

Why not include references 19-21 alongside 16 when introducing surface sampling. Would be helpful to understand if they used circumferential or unidirectional probes.

Materials and Methods

I doubt you chose pigs because of the selection of sensor. Better to say that there is a wealth of experience and global acceptance of translational porcine models, and you used CE-marked human probes that could be rapidly translated to a clinical setting.

Ethics

No comment

Animals

Worth mentioning pentobarbital + morphine + KCl euthanasia at end of procedure.

Laparotomy

This is the first time you indicate that you wanted to do normal circulation experiments in the caput and ischemic experiments in the cauda. Is this a correct interpretation of the reason why you describe abandoning measurements in the caput?

Microdialysis catheters and pCO2 sensors

Did you use 4 MD probes per pig? Make sure this is reflected earlier in the Methods summary. Would be good to have an annotated photograph of the actual pancreas with probes in situ, and an illustration or table of probes and their placements.

From the second pig we additionally -> From the second pig onwards, in addition to

catheters, used -> catheters, we used

Is there a reason you chose HES as opposed to a simple crystalloid?

You can say microvials rather than small plastic bottles -> the microdialysate was collected in microvials (MDialysis AB, etc.)

I believe MDialysis refer to the Iscus Flex as the ISCUSflex or ISCUSflex (superscript)

No need to mention glycerol – cut this sentence

Experimental protocol

Remove second comma in first sentence.

Here is the first place you confirm all measurements were performed in the cauda. Make sure this aligns with the description in Laparotomy, or did these problems with the caput have no material impact on the study performance?

Arterial clamping was kept for -> The XX artery was kept clamped for

Measurements

What do you mean by additional registrations? How frequent were these?

When you say the pCO2 values were extracted with 5 minute intervals, do you mean that you averaged the reading of the preceding 5 minutes, or that you only took the sample that occurred on the 5 minute mark, or that you analysed every sample but downloaded them off the system every 5 minutes?

Statistical analysis

Please note here whether you received input from a statistician during analysis.

You mention that you did not perform a sample size calculation in the first sentence, followed by ‘for missing values when we determined the sample size’. Did you perform a sample size calculation, or do you mean that the data from this study could permit sample size calculations for subsequent studies?

Are you certain that your data (tissue lactate or pCO2 levels) are not normally distributed?

It would be interesting to record and remark on the deviation between two MD catheters or pCO2 probes in the same location, rather than assuming the mean value is the true result.

Results

You mention one surface catheter failed to sample after venous occlusion. Was this one of the two paired catheters, or one set of catheters from a single animal. Was the data kept or discarded?

It’s very interesting that there was no change in blood gas values during the experiment – perhaps worth a comment that this ‘classical test’ for organ ischaemia isn’t much use here. In fact, might be worth correlating ABG lactate against MD lactate.

Microdialysis

Lactate measured with MDialysis-65 -> Lactate measured with circumferential sampling microdialysis probes (you’ve already said which ones you’re using).

You call this section ‘circumferential sampling’ and mention the MDialysis-65 probe in the opening sentence. It is now a little confusing to understand the probes used in this experiment.

Up to now it appeared to be 2 of the same probe type placed on the surface or intraparenchymally, with the MDialysis-65 ones used intraparenchymally. Looking at table 3, did Pig 3 have 2 parenchymal MDialysis-65s, 1 surface MDialysis-65, and 1 surface OnZurf all placed and recording at the same time, or some other configuration? Please revisit the description in Method to make sure this is clear, perhaps with an illustration of probe types and placements, or table listing these.

It would also be helpful to understand why you would place a circumferential sampling MD probe on the surface of an organ – are there studies using the circumferential probe for surface measurements for example? Perhaps clarify the type of probes used in ref. 16 when this is referenced in Introduction.

You should mention arterial lactate here as well as venous.

Is there a value to discussing changes in glycerol if you cannot compare against the surface probes? Are you certain that the proximity of the OnZurf probe to the MDialysis-65 surface probe did not lead to contamination with glycerol?

Discussion

Delete first two commas in opening sentence.

Tissue-pCO2 -> tissue pCO2, no hyphen

In reference to lactate levels, please change ‘close correlation’ to ‘correlation’ in the first two sentences. Your R value for lactate was much lower than for pCO2 which did ‘correlate well’.

Why do you think you recorded higher surface lactate values than parenchymal except for episodes of arterial clamping? Was this for the OnZurf or the MDialysis probes? This should be discussed. The next paragraph does discuss the OnZurf, but the type of catheter for which you’re making this generalisation in the previous paragraph should be clear.

PCO2 (capital) -> Tissue pCO2

Conclusion

You can be stronger in your conclusions. They are both sensitive to changes in common ischemic markers, more so than venous or arterial sampling (standard clinical practice)

pCO2-sensors -> pCO2 sensors (no hyphen)

Clinical pancreas monitoring -> pancreatic monitoring in a clinical setting.

Reviewer #2: This is an interesting and novel paper describing the use of a diagnostic technique in an in-vivo porcine model.

Microdialysis has been described in various clinical, pharmacological and experimental settings. In the realm of organ transplantation it has been used in attempts to characterise organ injury and viability in in-vivo experimental models, clinical transplant models, and in conjunction with ex-vivo organ perfusion.

There is a growing recognition of the need to develop accurate, reliable and interpretable markers of organ viability – especially with the emergence of machine organ perfusion (cold or warm) as viable tool in organ preservation.

The group is from a large unit, that has published extensively on micro dialysis in animal and clinical models.

This paper investigates the potential for use of microdialysis assessment in pancreatic ischaemia using a porcine in-vivo model. One concern in the use of microdialysis in pancreatic surgery is the need for intra-parenchymal placement of the probe – which in liver or kidney settings is acceptable risk, but which in the pancreas disrupts the capsule, and fragile parenchymal with potential for pancreatitis, leak, or fistula. The experiment investigated the feasibility of using MD probes on the surface of the pancrease (to detect a range of metabolites), compared to traditional intra-parenchymally placed probes, as well as a dedicated designed surface MD probe, in a model of arterial and venous ischaemic injury in a porcine in-vivo model. In-vivo porcine pancreatic experimentation is notoriously difficult.

They report results for 8 pigs in these experiments. Overall the manuscript is well written.

Main questions / comments for author review:

1) Did the authors use any anticoagulation in the experiments?

2) Did the authors perform any histological analysis of the pancreatic tissue exposed to the arterial and venous ischaemia? Was there any evidence of microvascular compromise? Thrombosis? – or any macrovascular effect of the insults? – if not what was the reason for omitting this.

3) What were the sites of placement of the probes in the pancreatic tail– were the probes placed in the same location in each experiment? (A photo – of a pancreas in situ with the probes in place would be helpful to appreciate the described methods).

Were the surface probes placed on the pancreatic ‘capsule’ – or where they attempted to be place beneath this?

4) It is appreciated the authors used the same area of tissue as a control (as described in the methods)– pre/post vascular occlusion. However did they successfully measure changes or levels in the head of the pancreas during the clamping period as a concurrent control? It is unclear as the methods alude to this being attempted – in several pigs. What was the problem encountered in performing MD on tissue in the pancreatic head?

5) What is the surface area of the OnZurf probes and expected area that the probes can detect metabolites from? – a photo of both types of probes would be useful for readers. The authors also mention that surface probes detected lower levels in general than the parenchymally placed probes – and mention odema as potential factor – was odema a significant finding during the experiments - and at what time point i.e. before or after venous occlusion – are they any photos of a pancreas at various time points of the experiments. The authors should elaborate on the potential reasons for the differences between deep and surface levels.

6) It is appreciated this is a small feasibility study – it would be interesting to note if metabolite levels are different depending on the area of the pancreas sampled – between different deep parenchymal and surface methods (i.e. were the deep probes placed in the same area of the pancreas – and did the levels detected by each correlate, or were they different? – and the same for the surface probes? - The authors mention in the discussion about the larger variations between CO2 readings in the surface probes for example – but this is not described in the results.

7) It is not clear what the range of capabilities are for the OnZurf probe to detected different metabolites? The authors mention glycerol cannot be detected – are there any other restrictions?

8) In the 3 animals that failed to demonstrate venous ischaemia – what was the failed venous occlusion? In those animals that venous occlusion did appear to work - Was there any evidence of vascular thrombosis – especially towards the end of the procedure? And what observations made the authors determine venous occlusion failed?

9) The use of CO2 as a marker of ischaemia is noteworthy – have the authors thought about any additional markers of ischaemia that could be measured? Did the authors assess systemic markers of pancreatic injury? Amylase/lipase?

10) The discussion is appropriate and places the results and fair conclusions in the context of the objectives of the project. However further discussion on the next steps of this work – in relation to 1) experimental plans/aims or 2) clinical strategies to employ these techniques would be useful for reader especially placing this report in the context of their overall work.

11) Limitations are mentioned and relevant. The inability to have a concurrent non-ischaemic tissue control however is important – and perhaps the reasons behind this could be elaborated upon and potentially steps to mitigate against this in future.

12) The conclusion is very short – and would benefit from the points mentioned in (10) above – context of the work in terms of future research plans, and where the authors see these techniques leading.

6. PLOS authors have the option to publish the peer review history of their article (what does this mean?). If published, this will include your full peer review and any attached files.

Reviewer #1: **Yes: **Dr. Robert M. Learney PhD MRCS

Reviewer #2: No

---

## [Author Response · Author response to Decision Letter 0]

20 Dec 2021

Response to reviewers: Please view the Response to reviewers as attached file where in some of the answers pictures and graphs are included

The questions/comments are numbered 1:1-42 for reviewer 1 and 2:1-12 for reviewer 2 and are included in the rebuttal letter followed by our Response, and Manuscript changes. 

Reviewer #1: This reviewer has no concerns about research ethics or publication ethics.

It is a very interesting paper and adds to the literature around real-time microdialysis and gas sampling in organ transplantation.

1.1: I think the team have missed a potentially impactful aspect from their discussion, as this paper introduces the idea that microdialysis + pCO2 both gave more sensitive, more timely, and truly dynamic indications of tissue ischemia than the current clinical practices of venous and arterial lactate monitoring.

1.1: Response: We have highlighted this aspect of the paper by adding the following sentence to the discussion. 

1.1: Manuscript changes: The successful biosensor based local detection of pancreatic ischemia, together with the absence of systemic markers of ischemia supports previous findings, and emphasizes the need for local monitoring 1, 2

Specific comments by section:

Background

1.2: Pancreatic transplantation instead of pancreas transplantation

1.2: Response: We have replaced pancreas with pancreatic.

1.3: No need for hyphens in tissue-pCO2-sensors, just tissue pCO2 sensors. Same throughout manuscript: pCO2-sensors can just be pCO2 sensors.

1.3: Response: We have removed hyphens throughout the manuscript. 

Methods

1.4: Unclear which were inserted into parenchyma and which attached to surface of pancreas. Currently reads as though the MD catheter was inserted and the tissue pCO2 sensor was attached to the surface. Maybe “In 8 anaesthetized pigs, pairs of lactate monitoring microdialysis catheters and tissue pCO2 sensors were simultaneously inserted into the parenchyma and attached to the surface of the pancreas.”?

1.4: Response: The suggested improvement of the formulation was used as above. Indeed, both microdialysis catheters and tissue pCO2 sensors were used simultaneously in the parenchyma and on the surface.

1.4: Manuscript changes: In 8 anaesthetized pigs, pairs of lactate monitoring microdialysis catheters and tissue pCO2 sensors were simultaneously inserted into the parenchyma and attached to the surface of the pancreas.

1.5: No need for hyphens in tissue-pCO2-sensors, just tissue pCO2 sensors

PCO2 was measured…-> Tissue pCO2 was measured…(no capital letter P)

Results

PCO2 increased -> Tissue pCO2 increased

Conclusion -> Conclusions

pCO2-sensors -> tissue pCO2

1.5: Response: The hyphens have been removed accordingly in the manuscript.

1.6: Introduction

Pancreas transplantation -> pancreatic transplantation

1.6: Response: It has been corrected accordingly. 

1.7: Would be nice to give the percentage lost to thrombosis – it’s shockingly high

1.7: Response: The reported incidence of thrombosis varies a lot though, from 1-40%, probably because sometimes non-occlusive thrombi are reported in addition to occlusive, sometimes only thrombosis leading to graft losses are reported. Grafts lost after pancreas thrombosis has been reported to about 4-8% after SPK and 10-12% after solitary pancreatic transplantations3. In our center, we had 25% incidence of graft thrombosis between 2015-2018, of which 6% graft thrombosis lead to early graft loss4. It is generally agreed that graft thrombosis is the major cause of early graft loss after a pancreatic transplantation5-7. To give an idea of the numbers the following sentence has been added to the introduction:

1.7: Manuscript changes: Early graft thrombosis leads to graft loss in about 5-10%, and accounts for almost one third of grafts lost within the first 6 months after transplantation3, 6. 

1.8: The microdialysis catheters -> Microdialysis catheters (talking about the general item)

1.8: Response: It has been corrected.

1.9: Need manufacturer of pCO2 sensor

1.9: Response: The manufacturer of the IscAlertTM is Sensocure (https://sensocure.no/).The full reference is now added to the Introduction in addition to the Methods section. 

1.9: Manuscript changes: The tissue pCO2 sensor (IscAlertTM, Sensocure AS, Skoppum, Norway) is based on the principle of CO2-conductivity

1.10: In addition to ref. 15, consider https://doi.org/10.1039/C8AY01807C

1.10: Response: The use of microdialysis for organ monitoring ex vivo as part of viability testing of the organ before transplantation is a very interesting field of application of organ close monitoring techniques. The reference has been added and the sentence in the Introduction changed to:

1.10: Manuscript changes: Microdialysis catheters positioned in the parenchyma have confirmed pancreatic ischemia in a porcine model in vivo 2, in porcine and human pancreases ex vivo8, and we have monitored pancreas grafts with microdialysis catheters on the organ surface 1. 

1.11: Why not include references 19-21 alongside 16 when introducing surface sampling. Would be helpful to understand if they used circumferential or unidirectional probes.

1:11: Response: We agree that the Introduction would benefit from a clarification of which probes –circumferential or unidirectional- were used and a better introduction to the unidirectional sampling probe. We have clarified the type of probe used, and added the suggested references in the introduction with the following sentence: 

1.11: Manuscript changes: In this study conventional circumferential microdialysis probes, designed to be inserted into the tissue, were used on the organ surface. To customize probes for surface sampling, a unidirectional sampling microdialysis probe (OnZurf Probe, Senzime AB, Uppsala, Sweden) has been developed and tested on small bowel and esophagus 9, 10

1.11: Response: Since this information now appears in the introduction, we have removed the following sentence from the discussion: 

1.11: Manuscript changes: The OnZurf catheter has successfully detected ischemia on the organ surfaces of small bowel and esophagus in animal models 9, 10.

Materials and Methods

1.12: I doubt you chose pigs because of the selection of sensor. Better to say that there is a wealth of experience and global acceptance of translational porcine models, and you used CE-marked human probes that could be rapidly translated to a clinical setting.

1.12: Response: We have changed the sentence in the Materials and methods section to the following: 

1.12: Manuscript changes: The porcine model was chosen since there is a wealth of experience and global acceptance of translational porcine models. We used CE-marked human microdialysis probes and tissue pCO2 sensors that are currently used in clinical studies, and in the process of receiving a CE marking, that could rapidly be adapted to a clinical setting.

Ethics

No comment

Animals

1.13: Worth mentioning pentobarbital + morphine + KCl euthanasia at end of procedure.

1.13: Response: We have now added the euthanasia procedure into the text as well, in addition to the supplementary table. 

1.13: Manuscript changes: At the end of the procedure, the animals were euthanized with pentobarbital (1000 mg), morphine (50-100 mg) and potassium chloride (70-100 mmol).

Laparotomy

1.14: This is the first time you indicate that you wanted to do normal circulation experiments in the caput and ischemic experiments in the cauda. Is this a correct interpretation of the reason why you describe abandoning measurements in the caput?

1.14: Response: Yes, we described this attempt of keeping the caput non-ischemic to explain that we aimed for a non-ischemic control tissue. However, to ascertain ischemia in the cauda, we did not succeed keeping the caput non-ischemic and had to abandon this set-up. Instead, we chose to use baseline values from the sensors in the pancreatic tail before inducing ischemia, as a non-ischemic reference. We have clarified this by making changes under the Laparotomy section, with the following wording: 

1.14: Manuscript changes: In the four first animals, the pancreas was divided to isolate the corpus and the cauda on each side of the portal vein, aiming to keep normal circulation in the caput while performing ischemic experiments on the cauda. However, clamping of the vessels to the cauda also induced ischemia in the caput in the first four animals, and these measurements could therefore not serve as non-ischemic controls and were discarded from the study. Instead, the baseline measurements before occlusion, were used as non-ischemic reference values.

Microdialysis catheters and pCO2 sensors

1.15: Did you use 4 MD probes per pig? Make sure this is reflected earlier in the Methods summary. Would be good to have an annotated photograph of the actual pancreas with probes in situ, and an illustration or table of probes and their placements.

1.15: Response: Number of catheters used in every location is now displayed in a supplementary S3 Table and a photograph of the set up in S1 Fig. To clarify this, the text has also been changed in the section “Microdialysis catheters and tissue pCO2 sensors” and the wording is now the following: 

1.15: Manuscript changes: We used two catheters/sensors of every type in the parenchyma and two on the surface to assure measurements from every location, except for one animal with no and three animals with only one unidirectional sampling microdialysis and two animals with only one surface pCO2 sensor. Number of catheters/sensors placed in the parenchyma and on the surface of every pig is displayed in S3 Table and a picture of the set-up in S1 Fig. Circumferential sampling microdialysis catheters with a 30 mm membrane, a pore size of 100 kDa, and an outer diameter 0.6 mm (MDialysis-65 catheter, M Dialysis AB, Stockholm, Sweden) were inserted into the pancreatic parenchyma using splittable introducer needles and placed onto the pancreatic capsule and sutured to the surface of the pancreas. Tissue pCO2 sensors (IscAlertTM, Sensocure AS, Skoppum, Norway), with diameter 0.7 mm, were inserted into the pancreatic tissue and sutured on the surface in the same manner as with the microdialysis catheters.

1.16: From the second pig we additionally -> From the second pig onwards, in addition to

catheters, used -> catheters, we used

1.16: Response: The sentence has been corrected accordingly.

1.17: Is there a reason you chose HES as opposed to a simple crystalloid?

1.17: Response: In our study we used circumferential sampling microdialysis catheters with 100 kDa pore membranes. For these catheters our research group performed a study examining seven different perfusion fluids with respect to recovery of metabolic and inflammatory markers11. The two crystalloids tested showed no fluid recovery, probably due to leakage into the surroundings through the semipermeable membrane, and crystalloids were therefore not recommended. The mentioned study concluded that HES 130/0.4 could replace the dextran 60 that had been recommended until then, and therefore we use HES 130/0.4 as a standard microdialysis perfusion fluid in our experiments. 

1.18: You can say microvials rather than small plastic bottles -> the microdialysate was collected in microvials (MDialysis AB, etc.)

I believe MDialysis refer to the Iscus Flex as the ISCUSflex or ISCUSflex (superscript)

1.18: Response: We have changed these words accordingly.

1.19: No need to mention glycerol – cut this sentence

1.19: Response: This sentence has been cut. 

Experimental protocol

1.20: Remove second comma in first sentence.

1.20: Response: The punctuation has been corrected accordingly. 

1.21: Here is the first place you confirm all measurements were performed in the cauda. Make sure this aligns with the description in Laparotomy, or did these problems with the caput have no material impact on the study performance?

1.21: Response: The cauda was chosen for ischemic measurements since it was easier to control the blood flow to this part of the pancreas, while more arteries and branches of arteries supplied the caput. As explained in response 1:14, we did not manage to keep the caput non-ischemic and baseline measurements from the cauda were used as non-ischemic reference. We consider this was a valid replacement as a reference, and we do not consider this had any negative impact on the study performance. As already referred to in response 1:14 we have changed the text under “Laparotomy” to more clearly explain what the original idea was and which were the measurements used. 

1.22: Arterial clamping was kept for -> The XX artery was kept clamped for

1.22: Response:”Arterial clamping was kept…” has been changed to: 

1.22: Manuscript changes: “The coeliac trunk and the proper hepatic artery were clamped…”

Measurements

1.23: What do you mean by additional registrations? How frequent were these?

1.23: Response: We monitored heart rate, blood pressure, CVP and SpO2 continuously and recorded the values every 15 minutes. Additional registrations were made occasionally and were not used in the calculations. Therefore, we consider that this sentence is unnecessary, and we have removed it from the manuscript. 

1.24: When you say the pCO2 values were extracted with 5 minute intervals, do you mean that you averaged the reading of the preceding 5 minutes, or that you only took the sample that occurred on the 5-minute mark, or that you analysed every sample but downloaded them off the system every 5 minutes?

1.24: Response: PCO2 was recorded 1-2 times per second, but only recordings on every 5 minute mark were used for calculations. We have changed the sentence under “Measurements” to the following: 

1.24: Manuscript changes: Only the pCO2-values occurring on every 5-minute mark were used for calculations. 

Statistical analysis

1.25: Please note here whether you received input from a statistician during analysis.

1.25: Response: The statistical analyses were reviewed by a statistician. The following sentence has been added to the “Statistical analysis” section: 

1.25: Manuscript changes: The statistical analyses were reviewed by Øivind Skare, statistician and researcher at the National Institute of Occupational Health Oslo, Norway.

1.26: You mention that you did not perform a sample size calculation in the first sentence, followed by ‘for missing values when we determined the sample size’. Did you perform a sample size calculation, or do you mean that the data from this study could permit sample size calculations for subsequent studies?

1.26: Response: We performed a power calculation, assuming a minimum lactate increase of 50%, a power of 80% and a significance level of 5%, suggest a minimal sample size of 4. Since the experimental model was new and first needed to be tested with pilot animals and we needed to account for missing samples, we ended up with eleven animals. We have changed one sentence and added one sentence to explain this under “Statistical analysis”: 

1.26: Manuscript changes: Based on data from a previous study 12, we anticipated a more than 100% increase in the main target variables, lactate and pCO2, following total vessel occlusion. A power calculation assuming a minimum lactate increase of 50%, a power of 80% and a significance level of 5% suggested a minimal sample size of 4 animals.

1.27: Are you certain that your data (tissue lactate or pCO2 levels) are not normally distributed?

1.27: Response: We used histogram and Shapiro Wilks test to decide normality. Shapiro Wilks test for both CO2 and lactate returned p<0.001 as a sign of non-normality. Histograms displayed in Response to reviewers attachment. 

The following sentence is added under statistical analysis:

1.27: Manuscript changes: All variables apart from blood pressures were non-normally distributed according to histogram and Shapiro Wilk’s test. 

1.28: It would be interesting to record and remark on the deviation between two MD catheters or pCO2 probes in the same location, rather than assuming the mean value is the true result.

1.28: Response: We placed two catheters of the same type in each location mainly to assure samples from each location in case malfunction of one of the catheters. Wherever we had results from two catheters/sensors of the same type and location (parenchyma or surface), we chose the mean values of the two, since we for some of the locations only have results from one sensor or catheter. To test correlation between results from two catheters in the same position would be informative of the reliability of the methods. Therefore, we have now analyzed all time points where we have data from two probes placed in the same location, and added this information in two supplementary tables S4 Table (microdialysis) and S5 Table (tissue pCO2 sensors). 

In the “Statistical analysis” section: 

1.28: Manuscript changes: Pearson correlation coefficients were calculated for every pair of catheters or sensors placed in the same location.

In the result section-Microdialysis-Circumferential sampling:

1.28: Manuscript changes: Correlation coefficient (R) between pairs of catheters placed in parenchyma and surface are displayed in Table S4. 

In the result section-Microdialysis-unidirectional sampling:

1.28: Manuscript changes: Correlation coefficients (R) were calculated between pairs of unidirectional sampling catheters placed for four pigs (S4 Table).

In the results section PCO2:

1.28: Manuscript changes: Correlation coefficients between pairs of pCO2 sensors placed in the parenchyma (n=8) and on the surface (n=2) are displayed in S4 Table. 

Results

1.29: You mention one surface catheter failed to sample after venous occlusion. Was this one of the two paired catheters, or one set of catheters from a single animal. Was the data kept or discarded?

1.29: Response: In one animal, one set of catheters failed to sample: one catheter failed to sample during the whole experiment and one catheter sampled successfully during the arterial occlusion and then failed. The data was discarded. For clarification we changed the text under “Results”: 

1.29: Manuscript changes: In one animal, data from surface catheters are based on one catheter only during arterial occlusion and no data was obtained during venous occlusion due to malfunction of sensors.

1.29: Response: The information of number of sampling catheters from each location is added in the new supplementary S3 Table, as mentioned in response 1:15, and the following sentence has in addition been added to the “Results” section: 

1.29: Manuscript changes: The number of catheters/sensors with successful measurements from each location are displayed in S3 Table. 

1.30: It’s very interesting that there was no change in blood gas values during the experiment – perhaps worth a comment that this ‘classical test’ for organ ischaemia isn’t much use here. In fact, might be worth correlating ABG lactate against MD lactate.

1.30: Response: To highlight the comparison with lactate in venous and arterial blood, we refer to the change in the discussion mentioned in response 1:1, and in addition we rephrased the first sentence of the conclusion to the following: 

1.30: Manuscript changes: In conclusion, organ close microdialysis and tissue pCO2 sensors detected and demonstrated dynamic and timely indications of pancreas tissue ischemia, not reflected with the current practices of systemic arterial and venous lactate monitoring.

1.30: Response: Regarding correlating ABG lactate against MD lactate, we do not consider it important enough to include as supplementary data in the article. Since we had very little variation in lactate in the blood gases, it is expected not to correlate with the dynamic measurements of the microdialysis catheters. To demonstrate this, we refer to the figures and correlations in the table in the attachment file Response to reviewers. 

 Microdialysis parenchymal lactate: 

Arterial blood: R=0.0085 (p=0.30)

Venous blood: R=0.015 (p=0.15)

 Microdialysis surface (circumferential):

Arterial blood: R= -0.0063 (p=0.54)

Venous blood: R= 0.0085 (p=0.51)

 Microdialysis surface (unidirectional): 

Arterial blood: R= -0.027 (p=0.074)

Venous blood: R= -0.014 (p=0.45)

Microdialysis

1.31: Lactate measured with MDialysis-65 -> Lactate measured with circumferential sampling microdialysis probes (you’ve already said which ones you’re using).

1.31: Response: It has been changed accordingly. 

1.32: You call this section ‘circumferential sampling’ and mention the MDialysis-65 probe in the opening sentence. It is now a little confusing to understand the probes used in this experiment.

1.32: Response: We have now clarified which catheters were used by more consequently terming them circumferential or unidirectional sampling. 

1.33: Up to now it appeared to be 2 of the same probe type placed on the surface or intraparenchymally, with the MDialysis-65 ones used intraparenchymally. Looking at table 3, did Pig 3 have 2 parenchymal MDialysis-65s, 1 surface MDialysis-65, and 1 surface OnZurf all placed and recording at the same time, or some other configuration? Please revisit the description in Method to make sure this is clear, perhaps with an illustration of probe types and placements, or table listing these.

1.33: Response: We have clarified the number of catheters at each location and from how many catheters we have results by adding supplementary S3 Table. MDialysis-65 catheters were the only microdialysis catheters used in the parenchyma, while both MDialysis-65 and OnZurf catheters were used on the surface. Table 3 solely displays the correlations between measurements from the parenchyma and from the surface for both microdialysis catheters. For clarification we renamed MDialysis-65 circumferential and OnZurf unidirectional in Table 3. 

1.34: It would also be helpful to understand why you would place a circumferential sampling MD probe on the surface of an organ – are there studies using the circumferential probe for surface measurements for example? Perhaps clarify the type of probes used in ref. 16 when this is referenced in Introduction.

1.34: Response: Yes, exactly. The first studies evaluating microdialysis monitoring from organ surfaces were using conventional circumferential sampling catheters13, 14, and concluded that the surface measurements followed the parenchymal, but that these probes were not optimal for surface sampling. This group has then collaborated with the company Senzime (https://senzime.com/) to develop a probe more suitable for surface sampling, with the resulting Onzurf probe. On the Onzurf probe the tip of the catheter with the semipermeable membrane has been covered on three sides allowing unidirectional sampling and it can be easily attached by sutures to the organ surface. The round circumferential sampling probe is not so easily attached to the organ surface and the tip might point out from the surface and sample from the surroundings as well. We have clarified this in the introduction by adding: 

1.34: Manuscript changes: In these studies, where microdialysis catheters were first used on organ surfaces, conventional circumferential microdialysis probes, designed to be inserted into the tissue, were used on the organ surface. To customize probes for surface sampling, a unidirectional sampling microdialysis probe (OnZurf Probe, Senzime AB, Uppsala, Sweden) was developed and has been tested on small bowel and esophagus 9, 10

1.35: You should mention arterial lactate here as well as venous.

1.35: Response: We have added values from arterial lactate during the experiment to Table 2. The first sentence in the Results-microdialysis-circumferential sampling section has been changed to include arterial blood lactate: 

1.35: Manuscript changes: Lactate measured with circumferential sampling microdialysis probes increased in the pancreas parenchyma and on the organ surface in response to pancreatic ischemia, without changes in systemic arterial or venous blood lactate

1.36: Is there a value to discussing changes in glycerol if you cannot compare against the surface probes? Are you certain that the proximity of the OnZurf probe to the MDialysis-65 surface probe did not lead to contamination with glycerol?

1.36: Response: Glycerol is a marker of cell membrane degradation, and thereby serves as a marker of damage. Therefore, we believe it is an important marker to include in this study on organ ischemia and we consider it relevant to compare glycerol in the parenchyma and on the surface. It is only with the Onzurf probe (unidirectional) that glycerol cannot be measured, since the membrane of these catheters is coated with glycerol. We regard this is a limitation of the current Onzurf catheters. 

We are not concerned that there was any contamination of glycerol from the Onzurf catheters to the surrounding. Even if theoretically possible, we consider this is unlikely since glycerol measured with the MDialysis catheters on the organ surface was stable before induction of ischemia and then followed the dynamic course of the ischemia and reperfusion protocol. The glycerol measured from the Onzurf catheters demonstrated initially very high values, and then we saw a gradual wash-out during the experiment. The inserted picture, in the attachment file Response to reviewers, shows an example from one of the pigs, with the green curves representing glycerol measured with the Onzurf catheter. 

Discussion

1.37: Delete first two commas in opening sentence.

Tissue-pCO2 -> tissue pCO2, no hyphen

1.37: Response: These have been changed accordingly. 

1.38: In reference to lactate levels, please change ‘close correlation’ to ‘correlation’ in the first two sentences. Your R value for lactate was much lower than for pCO2 which did ‘correlate well’.

1.38: Response: It has been changed accordingly. 

1.39: Why do you think you recorded higher surface lactate values than parenchymal except for episodes of arterial clamping? Was this for the OnZurf or the MDialysis probes? This should be discussed. The next paragraph does discuss the OnZurf, but the type of catheter for which you’re making this generalisation in the previous paragraph should be clear.

1.39: Response: To clarify which probes were discussed, we have rearranged this paragraph in the discussion so that circumferential sampling MDialysis probes are commented first and unidirectional sampling Onzurf catheters thereafter. The higher surface lactate values were observed with the MDialysis probes, which we assume was due to lactate accumulation in the surrounding abdominal fluid, detected by the circumferential sampling probe, as opposed to the Onzurf catheter. 

1.39: Manuscript changes: We used two different types of microdialysis catheters for the surface measurements: a traditional circumferential sampling microdialysis catheter (MDialysis-65) and a novel unidirectional sampling microdialysis catheter (OnZurf). The correlation between lactate on the pancreas surface and in the parenchyma, as observed with the circumferential sampling microdialysis catheters, is in line with a previous experimental study on liver where surface lactate concentrations reflected the parenchymal 14. In this previous study, the highest lactate values were detected on the organ surface during the ischemic episodes. In contrast, we measured higher peak lactate concentrations in the parenchyma during arterial ischemia, but similarly higher surface than parenchymal values during the rest of the experiment. Higher lactate values is to be expected in the parenchyma, since lactate is produced by the ischemic tissue. The higher surface lactate values, can be explained by lactate accumulation in the abdominal fluid, detected by the circumferential catheters that allow sampling from the surroundings. Probably due to the same reason we observed that surface lactate measured with the circumferential sampling catheters did not return to baseline after the first reperfusion. Conversely, lactate measured with unidirectional catheters returned to baseline. This could support the theory that these catheters are less influenced by the surroundings and that unidirectional sampling catheters, being encapsulated on three sides and only sampling towards the organ surface, is an advantageous alternative for organ surface monitoring.

1.40: PCO2 (capital) -> Tissue pCO2

1.40: Response: It has been corrected. 

Conclusion

1.41: You can be stronger in your conclusions. They are both sensitive to changes in common ischemic markers, more so than venous or arterial sampling (standard clinical practice)

1.41:Response: We have rephrased the conclusion in an attempt to reflect more aspects of our results: 

1.41:Manuscript changes: In conclusion, organ close microdialysis and tissue pCO2 sensors detected and demonstrated dynamic and timely indications of pancreatic tissue ischemia not reflected with the current practices of systemic arterial and venous lactate monitoring. Both techniques are promising for rapid detection of postoperative complications leading to organ ischemia. Since lactate and pCO2 detected ischemia with adequate correlation between parenchyma and surface of the pancreas, surface probes seem to be a promising option for monitoring in a clinical setting. 

1.42: pCO2-sensors -> pCO2 sensors (no hyphen)

Clinical pancreas monitoring -> pancreatic monitoring in a clinical setting.

1.42: Response: These have been revised. 

Reviewer #2: This is an interesting and novel paper describing the use of a diagnostic technique in an in-vivo porcine model.

Microdialysis has been described in various clinical, pharmacological and experimental settings. In the realm of organ transplantation it has been used in attempts to characterise organ injury and viability in in-vivo experimental models, clinical transplant models, and in conjunction with ex-vivo organ perfusion.

There is a growing recognition of the need to develop accurate, reliable and interpretable markers of organ viability – especially with the emergence of machine organ perfusion (cold or warm) as viable tool in organ preservation.

The group is from a large unit, that has published extensively on micro dialysis in animal and clinical models.

This paper investigates the potential for use of microdialysis assessment in pancreatic ischaemia using a porcine in-vivo model. One concern in the use of microdialysis in pancreatic surgery is the need for intra-parenchymal placement of the probe – which in liver or kidney settings is acceptable risk, but which in the pancreas disrupts the capsule, and fragile parenchymal with potential for pancreatitis, leak, or fistula. The experiment investigated the feasibility of using MD probes on the surface of the pancrease (to detect a range of metabolites), compared to traditional intra-parenchymally placed probes, as well as a dedicated designed surface MD probe, in a model of arterial and venous ischaemic injury in a porcine in-vivo model. In-vivo porcine pancreatic experimentation is notoriously difficult.

They report results for 8 pigs in these experiments. Overall the manuscript is well written.

Main questions / comments for author review:

2.1) Did the authors use any anticoagulation in the experiments?

2.1: Response: No anticoagulation was used in the experiments. 

2.2) Did the authors perform any histological analysis of the pancreatic tissue exposed to the arterial and venous ischaemia? Was there any evidence of microvascular compromise? Thrombosis? – or any macrovascular effect of the insults? – if not what was the reason for omitting this.

2.2: Response: We chose not to perform any histological analysis. We considered the risk of causing damage to the pancreas by performing biopsies during the different phases of the experimental protocol was too high. The arterial occlusion rendered the pancreas duskier in color, but the color changes were reversed after reperfusion. However, after the venous occlusion, we observed clear macroscopic effects of the pancreas, which was edematous, and the color changed to dark brown/purple/black. These changes were not reversible during the reperfusion period of two hours. Pictures of the pancreas before ischemic insults and after venous occlusion are included in the attachment file Response to reviewers:

Pancreas before ischemic insults Pancreas after venous occlusion of 45 minutes

2:3) What were the sites of placement of the probes in the pancreatic tail– were the probes placed in the same location in each experiment? (A photo – of a pancreas in situ with the probes in place would be helpful to appreciate the described methods).

Were the surface probes placed on the pancreatic ‘capsule’ – or where they attempted to be place beneath this?

2:3: Response: In most pigs we placed 6 probes on the surface and 4 probes in the parenchyma in the pancreatic tail. Exact number of probes placed in each pig is shown in Table S3. We placed probes both on the dorsal and ventral side of the pancreatic tail. 

2:4) It is appreciated the authors used the same area of tissue as a control (as described in the methods)– pre/post vascular occlusion. However did they successfully measure changes or levels in the head of the pancreas during the clamping period as a concurrent control? It is unclear as the methods alude to this being attempted – in several pigs. What was the problem encountered in performing MD on tissue in the pancreatic head?

2.4: Response: We refer to question 1:14 for response and manuscript changes.

2.5) What is the surface area of the OnZurf probes and expected area that the probes can detect metabolites from? – a photo of both types of probes would be useful for readers. The authors also mention that surface probes detected lower levels in general than the parenchymally placed probes – and mention odema as potential factor – was odema a significant finding during the experiments - and at what time point i.e. before or after venous occlusion – are they any photos of a pancreas at various time points of the experiments. The authors should elaborate on the potential reasons for the differences between deep and surface levels.

2.5: Response: According to the manufacturer the Onzurf has a surface area of 0.2 cm2. We have mentioned that the absolute levels detected with the Onzurf (unidirectional) probes from the surface were lower than those detected with circumferential sampling probes, which we speculate could be due to different membrane properties and recovery rates of the catheters. We believe this is a more plausible reason than edema, since the measurements from the Onzurf catheters were consistently lower all through the experiments compared to both parenchymal and surfal placement of the MDialysis probes (Fig 3), both before and after edema developed in the pancreas. In fact, we did not mention edema as a reason for the absolute lower levels. Edema was indeed a significant finding after venous occlusion. A picture of the pancreas after venous occlusion is included in response 2:2. We have added the membrane area of the Onzurf probe in the manuscript under “Microdialysis catheters and tissue pCO2 sensors” and included pictures of the probes in S2 Fig. 

2.5: Manuscript changes: “…we used unidirectional sampling microdialysis catheters with a 15 mm membrane, surface area of 0.2cm2 and a pore size of 10 kDa,…”

2.5: Manuscript changes: Pictures of the probes available in S2 Fig.

2.6) It is appreciated this is a small feasibility study – it would be interesting to note if metabolite levels are different depending on the area of the pancreas sampled – between different deep parenchymal and surface methods (i.e. were the deep probes placed in the same area of the pancreas – and did the levels detected by each correlate, or were they different? – and the same for the surface probes? - The authors mention in the discussion about the larger variations between CO2 readings in the surface probes for example – but this is not described in the results.

2.6: Response: We refer to question 1:28 for response and manuscript changes regarding correlations between probes placed in the same location. We did not systematically place the probes in the same location in every pig. In some pigs we placed catheters of the same type on the dorsal and frontal side, while in some pigs the catheters of the same type were placed closer to each other on the same side. It was for pragmatic reasons to fit all our probes. 

It is true that the larger variation in the CO2 sampling surface probes were not mentioned in the results and that it is not so easily accessible from Figure 4, where an idea of the spread of the results appear. We added the following sentence to the Results-PCO2 section: 

2.6: Manuscript changes: There were larger variations in the CO2 values and more surface sensors that failed to sample (S3 Table).

2.7) It is not clear what the range of capabilities are for the OnZurf probe to detected different metabolites? The authors mention glycerol cannot be detected – are there any other restrictions?

2.7: Response: Several factors affect the degree of recovery of different molecules when sampling with microdialysis. Such factors include molecular weight, diffusion coefficient, physical and chemical properties of the molecules and degree of hydrophilicity15. We have information from the producer that the membrane of the Onzurf catheter is comparable to that of the conventional microdialysis catheter, apart from a coating of glycerol. Importantly, the Onzurf catheter has a pore size of only 10kDa, restricting its ability to sample larger molecules such as inflammatory markers15. 

2.8) In the 3 animals that failed to demonstrate venous ischaemia – what was the failed venous occlusion? In those animals that venous occlusion did appear to work - Was there any evidence of vascular thrombosis – especially towards the end of the procedure? And what observations made the authors determine venous occlusion failed?

2.8: Response: The earliest sign of ischemia was from our organ close probes, both microdialysis and pCO2 sensors. However, we also noted a discoloration of the pancreas, which after venous occlusion got dark brown/purple/black together with an organ edema. Pictures are provided in response 2.2. Lack of any of these observations after clamping of the splenic vein determined failed venous occlusion. We have added the following sentence under “Results”: 

2.8: Manuscript changes: In three of the animals, clamping of the splenic vein did neither lead to any increased lactate or tissue pCO2, nor to any discolored or edematous pancreas, which we interpreted as failed venous occlusion. .

2.9) The use of CO2 as a marker of ischaemia is noteworthy – have the authors thought about any additional markers of ischaemia that could be measured? Did the authors assess systemic markers of pancreatic injury? Amylase/lipase?

2.9: Response: Regarding locally measured ischemia markers, previous similar studies investigating local ischemia have also used tissue pO2 and pH as indicators of ischemia 2, 16. In this study, we limited the use of sensors to two different types. 

In this experimental study, we did not assess the pancreas enzymes amylase and lipase from the systemic circulation. However, in a clinical study of pancreas graft ischemia there were no statistically significant differences neither for serum nor drain amylase when comparing patients with an early venous thrombi to those with no complications1. Likewise, after 40 minutes of pancreas ischemia in a porcine model, s-amylase did not increase2. These previous findings might suggest that s-amylase is not a reliable marker for early detection of a venous thrombosis or ischemia. Lipase would not be a suitable marker to follow a dynamic course due to its half-life of 10-12 hours17.Regarding its role in early detection of ischemia, it has to our knowledge not been investigated. 

2.10) The discussion is appropriate and places the results and fair conclusions in the context of the objectives of the project. However further discussion on the next steps of this work – in relation to 1) experimental plans/aims or 2) clinical strategies to employ these techniques would be useful for reader especially placing this report in the context of their overall work.

2. 10: Response: For the most reliable measurements parenchymal probes would probably be preferable, but the safety issue of inserting small sensors into the pancreatic parenchyma must first be resolved. More experimental studies are needed. Regarding surface monitoring, the influence of the probes from the surrounding and attachment to the organ surface stand out as the main challenges1. To solve this, the unidirectional sampling microdialysis probe is inviting. However, concerns can be raised if several sutures to the pancreas surface are safe. Whether tissue pCO2 sensors are similarly affected by e.g. hematomas as microdialysis catheters remain to be investigated. We have added the following section to the discussion: 

2.10: Manuscript changes: Monitoring of human pancreas with sensors inserted into the organ or attached onto the organ, needs to be preceded by a safety assessment study. One option could be monitoring animals for several days and a final examination of the pancreas histology regarding inflammation and leakages. However, since ischemia was accurately detected by surface sensors, a reasonablenext step would be to develop and improve surface monitoring. Unidirectional microdialysis is inviting, as presumed less influencde by the surroundings, but the current affixment to the organ surface needs to be evaluated from a safety perspective. Future studies are needed to explore if surface pCO2 sensors potentially can be influenced by e.g. a surrounding hematoma or a pancreatic leakage. 

2.11) Limitations are mentioned and relevant. The inability to have a concurrent non-ischaemic tissue control however is important – and perhaps the reasons behind this could be elaborated upon and potentially steps to mitigate against this in future.

2.11: Response: After splitting the pancreas in two parts we presumed that the head of the pancreas would have its blood supply from branches of the superior mesenteric artery, and that we could induce ischemia in the tail by clamping of the splenic artery. However, we realized in the pilot animals that there were other branches of the coeliac trunk supplying the body/tail, and therefore to ascertain ischemia in the body/tail we needed to clamp the coeliac trunk. 

However, clamping the coeliac trunk, also induced ischemia in the pancreatic head.Therefore, we were not able to achieve to ascertain ischemia in the tail of the pancreas without also inducing ischemia to varying degrees in the caput. Possibly clamping of the splenic artery would have been sufficient to induce ischemia in a more distal part of the pancreatic tail, and if clamping of the coeliac trunk was avoided the blood supply to the head would have been spared. However, with the number of sensors we used it would not have been practicable, due to the relatively small size of the tail. Another suggestion to achieve and ascertain both one ischemic part and one non-ischemic part of the pancreas could be to place a bypass from the aorta to the common hepatic artery and maintain blood supply to the caput via the gastroduodenal artery. We have added the following section to the discussion: 

2.11: Manuscript changes: To ascertain ischemia in the caudal part of the pancreas we had to clamp the coeliac trunk. As a result, we also induced ischemia in the head of the pancreas and concluded that the head was dependent on blood supply from the coeliac trunk. If a smaller and more distal part of the pancreatic tail was used, it is possible that clamping only the splenic artery would have been sufficient. Another way to keep the head non-ischemic could be a bypass from the aorta to the common hepatic artery and maintain blood supply to the head via the gastroduodenal artery.

2. 12) The conclusion is very short – and would benefit from the points mentioned in (10) above – context of the work in terms of future research plans, and where the authors see these techniques leading.

2. 12: Response: We refer to comment 1.41 for response and manuscript changes regarding the conclusion. 

References: 

1. Kjøsen G, Rydenfelt K, Horneland R, et al. Early detection of complications in pancreas transplants by microdialysis catheters, an observational feasibility study. PLoS One. 2021;16(3):e0247615. doi:10.1371/journal.pone.0247615

2. Blind PJ, Kral J, Wang W, et al. Microdialysis in early detection of temporary pancreatic ischemia in a porcine model. Eur Surg Res. 2012;49(3-4):113-20. doi:10.1159/000343806

3. Hakeem A, Chen J, Iype S, et al. Pancreatic allograft thrombosis: Suggestion for a CT grading system and management algorithm. Am J Transplant. Jan 2018;18(1):163-179. doi:10.1111/ajt.14433

4. Lindahl JP, Horneland R, Nordheim E, et al. Outcomes in Pancreas Transplantation With Exocrine Drainage Through a Duodenoduodenostomy Versus Duodenojejunostomy. Am J Transplant. 2018;18(1):154-162. doi:10.1111/ajt.14420

5. Farney AC, Rogers J, Stratta RJ. Pancreas graft thrombosis: causes, prevention, diagnosis, and intervention. Curr Opin Organ Transplant. Feb 2012;17(1):87-92. doi:10.1097/MOT.0b013e32834ee717

6. Muthusamy AS, Giangrande PL, Friend PJ. Pancreas allograft thrombosis. Transplantation. Oct 15 2010;90(7):705-7. doi:10.1097/TP.0b013e3181eb2ea0

7. Ramessur Chandran S, Kanellis J, Polkinghorne KR, Saunder AC, Mulley WR. Early pancreas allograft thrombosis. Clin Transplant. May-Jun 2013;27(3):410-6. doi:10.1111/ctr.12105

8. Sally A, Hamaoui K, Vallant N, et al. An improved rapid sampling microdialysis system for human and porcine organ monitoring in a hospital setting. Analytical Methods. 2018;10(44):5273-5281. doi:10.1039/c8ay01807c

9. Akesson O, Abrahamsson P, Johansson G, Blind PJ. Surface microdialysis on small bowel serosa in monitoring of ischemia. J Surg Res. Jul 2016;204(1):39-46. doi:10.1016/j.jss.2016.04.001

10. Akesson O, Falkenback D, Johansson G, Abrahamsson P. Surface Microdialysis Detects Ischemia After Esophageal Resection-An Experimental Animal Study. J Surg Res. Jan 2020;245:537-543. doi:10.1016/j.jss.2019.07.060

11. Froud T, Ricordi C, Baidal DA, et al. Islet Transplantation in Type 1 Diabetes Mellitus Using Cultured Islets and Steroid-Free Immunosuppression: Miami Experience. Am J Transplant. 2005;5(8):2037-2046. doi:10.1111/j.1600-6143.2005.00957.x

12. Pischke SE, Tronstad C, Holhjem L, Line PD, Haugaa H, Tonnessen TI. Hepatic and abdominal carbon dioxide measurements detect and distinguish hepatic artery occlusion and portal vein occlusion in pigs. Liver Transpl. Dec 2012;18(12):1485-94. doi:10.1002/lt.23544

13. Abrahamsson P, Aberg AM, Johansson G, Winso O, Waldenstrom A, Haney M. Detection of myocardial ischaemia using surface microdialysis on the beating heart. Clin Physiol Funct Imaging. May 2011;31(3):175-81. doi:10.1111/j.1475-097X.2010.00995.x

14. Abrahamsson P, Aberg AM, Winso O, Johansson G, Haney M, Blind PJ. Surface microdialysis sampling: a new approach described in a liver ischaemia model. Clin Physiol Funct Imaging. Mar 2012;32(2):99-105. doi:10.1111/j.1475-097X.2011.01061.x

15. Waelgaard L, Pharo A, Tonnessen TI, Mollnes TE. Microdialysis for monitoring inflammation: efficient recovery of cytokines and anaphylotoxins provided optimal catheter pore size and fluid velocity conditions. Scand J Immunol. Sep 2006;64(3):345-52. doi:10.1111/j.1365-3083.2006.01826.x

16. Pischke SE, Hyler S, Tronstad C, et al. Myocardial tissue CO2 tension detects coronary blood flow reduction after coronary artery bypass in real-timedagger. Br J Anaesth. Mar 2015;114(3):414-22. doi:10.1093/bja/aeu381

17. Ismail OZ, Bhayana V. Lipase or amylase for the diagnosis of acute pancreatitis? Clin Biochem. 2017;50(18):1275-1280. doi:10.1016/j.clinbiochem.2017.07.003

---

## [Decision Letter · Decision Letter 1]

7 Jan 2022

Microdialysis and CO2 sensors detect pancreatic ischemia in a porcine model

PONE-D-21-33061R1

Dear Dr. Rydenfelt,

We’re pleased to inform you that your manuscript has been judged scientifically suitable for publication and will be formally accepted for publication once it meets all outstanding technical requirements.

Kind regards,

Frank JMF Dor, M.D., Ph.D., FEBS, FRCS

Academic Editor

PLOS ONE

Additional Editor Comments (optional):

Reviewers' comments:

Reviewer's Responses to Questions

**Comments to the Author**

1. If the authors have adequately addressed your comments raised in a previous round of review and you feel that this manuscript is now acceptable for publication, you may indicate that here to bypass the “Comments to the Author” section, enter your conflict of interest statement in the “Confidential to Editor” section, and submit your "Accept" recommendation.

Reviewer #1: All comments have been addressed

Reviewer #2: All comments have been addressed

2. Is the manuscript technically sound, and do the data support the conclusions?

Reviewer #1: Yes

Reviewer #2: Yes

3. Has the statistical analysis been performed appropriately and rigorously? 

Reviewer #1: Yes

Reviewer #2: Yes

4. Have the authors made all data underlying the findings in their manuscript fully available?

Reviewer #1: Yes

Reviewer #2: Yes

5. Is the manuscript presented in an intelligible fashion and written in standard English?

Reviewer #1: Yes

Reviewer #2: Yes

6. Review Comments to the Author

Reviewer #1: Thank you for the revisions. It reads very well and is a very interesting study.

There are some tiny changes you could make (M Dialysis -> MDialysis in one place, occasional double spacing) but overall this is a good paper.

Reviewer #2: Thank you for the opportunity to re-review this interesting article.

The authors have diligently responded to the comments from both reviewer 1 and reviewer 2, and have taken on board their suggestions.

Further comments:

2.5: The author response and changes made are reasonable and welcome. The original comment regarding ‘oedema’ was extrapolated from the authors comment in the discussion ‘’We speculate that this may be due to that some of the surface sensors were positioned under the organ surrounded by fluid, and others on the upper side of the organ occasionally towards air.’’ It is now clear that this explanation was unclear as to a possible cause of the discrepancies in levels detected between the surface and parenchymal probes. Changes in the revised manuscript and the responses to both reviewer 1 and reviewer 2 appear to have helped clarify this.

No further comments.

7. PLOS authors have the option to publish the peer review history of their article (what does this mean?). If published, this will include your full peer review and any attached files.

Reviewer #1: **Yes: **Dr. Robert M. Learney PhD MRCS

Reviewer #2: No

---

## [Editor Report · Acceptance letter]

2 Feb 2022

PONE-D-21-33061R1 

Microdialysis and CO_2_ sensors detect pancreatic ischemia in a porcine model 

Dear Dr. Rydenfelt:

I'm pleased to inform you that your manuscript has been deemed suitable for publication in PLOS ONE. Congratulations! Your manuscript is now with our production department. 

Kind regards, 

on behalf of

Dr. Frank JMF Dor 

Academic Editor

PLOS ONE